# Stateful active facilitator: Coordination and Environmental Heterogeneity in Cooperative Multi-Agent Reinforcement Learning

**Dianbo Liu**[*,1]      **Vedant Shah**[*,1,2]      **Oussama Boussif**[*,1]      **Cristian Meo**[3]

**Anirudh Goyal**[1]   **Tianmin Shu**[4]   **Michael Mozer**[5]   **Nicolas Heess**[6]   **Yoshua Bengio**[1,7]

## Abstract

In cooperative multi-agent reinforcement learning, a team of agents works together to achieve a common goal. Different environments or tasks may require varying degrees of coordination among agents in order to achieve the goal in an optimal way. The nature of coordination will depend on the properties of the environment—its spatial layout, distribution of obstacles, dynamics, etc. We term this variation of properties within an environment as *heterogeneity*. Existing literature has not sufficiently addressed the fact that different environments may have different levels of heterogeneity. We formalize the notions of *coordination level* and *heterogeneity level* of an environment and present **HECOGrid**, a suite of multi-agent RL environments that facilitates empirical evaluation of different MARL approaches across different levels of coordination and environmental heterogeneity by providing a quantitative control over coordination and heterogeneity levels of the environment. Further, we propose a Centralized Training Decentralized Execution learning approach called **Stateful Active Facilitator (SAF)** that enables agents to work efficiently in high-coordination and high-heterogeneity environments through a differentiable and shared knowledge source used during training and dynamic selection from a shared pool of policies. We evaluate SAF and compare its performance against baselines IPPO and MAPPO on HECOGrid. Our results show that SAF consistently outperforms the baselines across different tasks and different heterogeneity and coordination levels. We release the code for HECOGrid[1] as well as all our experiments.

## 1 Introduction

Multi-Agent Reinforcement Learning (MARL) studies the problem of sequential decision-making in an environment with multiple actors. A straightforward approach to MARL is to extend single agent RL algorithms such that each agent learns an independent policy (Tan, 1997). de Witt et al. (2020) recently showed that PPO, when used for independent learning in multi-agent settings (called Independent PPO or IPPO) is in fact capable of beating several state-of-the-art approaches in MARL on competitive benchmarks such as StarCraft (Samvelyan et al., 2019). However, unlike most single-agent RL settings, learning in a multi-agent RL setting is faced with the unique problem of changing environment dynamics as other agents update their policy parameters, which makes it difficult to learn optimal behavior policies. To address this problem of environment non-stationarity, a class of approaches called Centralized Training Decentralized Execution (CTDE) such as MADDPG (Lowe et al., 2017), MAPPO (Yu et al., 2021), HAPPO and HTRPO (Kuba et al., 2021) was developed. This usually consists of a centralized critic during training which has access to the observations of every agent and guides the policies of each agent. In many settings, MARL manifests itself in the form

---

[*]Equal Contribution, [1]Mila, Quebec AI institute, [2]BITS, Pilani, [3]TU Delft, [4]MIT, [5]Google Research, Brain Team, [6]Deepmind. [7]CIFAR Chair. Corresponding authors: liudianbo@gmail.com, vedantshah2012@gmail.com, oussama.boussif@mila.quebec

[1]https://github.com/veds12/hecogrid and https://github.com/jaggbow/saf

of cooperative tasks in which all the agents work together in order to achieve a common goal. This requires efficient coordination among the individual actors in order to learn optimal team behavior. Efficient coordination among the agents further aggravates the problem of learning in multi-agent settings.

| Benchmark | Cooperative | Partial Obs. | Image Obs. | Coordination Control | Heterogeneity Control |
|---|---|---|---|---|---|
| SMAC | ✓ | ✓ | × | × | × |
| MeltingPot | ✓ | ✓ | ✓ | ✓ | × |
| MPE | ✓ | ✓ | × | × | × |
| SISL | ✓ | × | × | × | × |
| DOTA 2 | ✓ | ✓ | × | × | × |
| HECOGrid | ✓ | ✓ | ✓ | ✓ | ✓ |

Table 1: Comparison between our newly developed HECOGrid environments and widely used multi-agent reinforcement learning environments including SMAC (Vinyals et al., 2017), MeltingPot (Leibo et al., 2021), MPE (Lowe et al., 2017), SISL (Gupta et al., 2017) and DOTA2 (Berner et al., 2019)

Another challenge in practical multi-agent learning problems is *heterogeneity* in the environment. Environment heterogeneity in reinforcement learning has previously been studied in the context of federated learning in Jin et al. (2022) which considers the problem of jointly optimizing the behavior of $n$ agents located in $n$ identical environments (same state space, same action space, and same reward function) with differing state-transition functions. However, in some real-world multi-agent learning problems, environment properties such as structure, dynamics, etc, may also vary within an environment, as compared to varying across different environments. Unmanned guided vehicles (UGVs) used for search and exploration may encounter different conditions, such as different distribution of obstacles or different terrains leading to differing dynamics in different regions. Warehouse robots coordinating to pick up a bunch of items might have to work in conditions varying from one section of the warehouse to another such as different organization of aisles. Similarly, as argued in Jin et al. (2022), an autonomous drone should be able to adapt to different weather conditions that it encounters during its flight. We build upon the formulation of Jin et al. (2022) to address the broader problem of heterogeneity within the environment.

We formally define two properties of an environment: **heterogeneity**, which is a quantitative measure of the variation in environment dynamics within the environment, and **coordination** which is a quantitative measure of the amount of coordination required amongst agents to solve the task at hand (we formally define **heterogeneity** and **coordination** in Section 3). The difficulty of an environment can vary based on the amount of heterogeneity and the level of coordination required to solve it. In order to investigate the effects of coordination and environmental heterogeneity in MARL, we need to systematically analyze the performance of different approaches on varying levels of these two factors. Recently, several benchmarks have been proposed to investigate the coordination abilities of MARL approaches, however, there exists no suite which allows systematically varying the heterogeneity of the environment. A quantitative control over the required coordination and heterogeneity levels of the environment can also facilitate testing the generalization and transfer properties of MARL algorithms across different levels of coordination and heterogeneity. A detailed analysis of the existing benchmarks can be found in Appendix A.1

Previous MARL benchmarks have largely focused on evaluating coordination. As a result, while, there has been a lot of work which attempts addressing the problem of coordination effectively, environment heterogeneity has been largely ignored. Heterogeneity so far has been an unintentional implicit component in the existing benchmarks. Hence, the problem of heterogeneity hasn't been sufficiently addressed. This is also apparent from our results where the existing baselines do not perform very competitively when evaluated on heterogeneity, since they were mainly designed to address the problem of coordination. Moreover, the fact that heterogeneity has been an unintentional implicit component of existing benchmarks, further strengthens our claim that heterogeneity is an essential and exigent factor in MARL tasks. Coordination and heterogeneity are ubiquitous factors for MARL. We believe that explicitly and separately considering these two as a separate factor and isolating them from other factors contributing to environment difficulty, will help motivate more research in how these can be tackled.

To address these limitations, we propose *HECOGrid*, a procedurally generated Multi-Agent Benchmark built upon MARLGrid (Ndousse et al., 2021). HECOGrid consists of three different environments which allow the testing of algorithms across different coordination and environmental heterogeneity levels. Each environment consists of $N$ agents and $M$ treasures, where the goal is to maximize the total number of treasures picked up by the agents in one episode. $c$ agents are required to pick up a single treasure, where $c$ is the coordination level of the environment. The environment is spatially divided into $h$ zones, and the environment dynamics vary from zone to zone. $h$ is the level of heterogeneity of the environment. $c$, $h$, $N$, and $M$ are controllable parameters. Table 1 presents a qualitative comparison between HECOGrid and other commonly used Multi-Agent RL environment suites.

HECOGrid also allows complete control over size of the map, number of obstacles, number of treasures, number of agents in addition to the coordination and heterogeneity levels. This provides ease of use of environments from small to very large scale with respect to the aforementioned parameters. This allows HECOGrid to be used as a standard challenging benchmark for evaluating not only coordination and heterogeneity but a lot of other factors. A lot of existing benchmarks (Vinyals et al., 2017; Samvelyan et al., 2019; Berner et al., 2019) focus on moving away from toy-like grid world scenarios, using more complex scenarios with high dimensional observation and action spaces, continuous control, challenging dynamics and partial observability. Although HECOGrid has partial and image observations, it has a relatively small and discrete action space. Hence, HECOGrid cannot be used to test how an algorithm fares in continuous control and high dimensional action space scenarios. However, in most of existing standard benchmarks, it is non-trivial to modify environment parameters and hence it is difficult to perform a wide range of generalization and robustness studies. Melting Point (Leibo et al., 2021) allows evaluating for out of distribution generalization, where the OOD scenarios can be defined by changing the *background population* of the environment. Unlike HECOGrid however, the physical layout of the environment (*substrate*) however, cannot be changed. HECOGrid, providing complete control over these environment parameters, make it easy to perform a wide range of experiments.

Further, to enable efficient training of MARL agents in high coordination and heterogeneity, we introduce a novel approach called **Stateful Active Facilitator** (SAF). SAF uses a shared *knowledge source* during training which learns to sift through and interpret signals provided by all the agents before passing them to the centralized critic. In this sense, the knowledge source acts as an information bottleneck and helps implement a more efficient form of centralization by refining the information being passed to the critic. Further, recent work in modular deep learning (Goyal et al., 2021a;d;b; Rahaman et al., 2021) has shown that different neural modules trained on a common objective lead to the emergence of specialist neural modules which help in improving performance via decomposition of the task. We hypothesize that a similar form of modularity can also be helpful in tackling the problem of heterogeneity. Instead of each agent using an individual monolithic policy, we propose the use of a *pool of policies* that are shared across different agents (Goyal et al., 2021a). At each time step, each agent picks one policy from the pool which is used to determine its next action where the selection being conditioned on the current state of the agent. During execution the parameters of the shared pool of policies can be distributed to each agent which can then operate in a completely decentralized manner. Hence our method falls under the umbrella of Centralized Training Decentralized Execution (CTDE) methods.

**Contributions.** We introduce a set of cooperative MARL environments with adjustable coordination and heterogeneity levels. We also propose SAF- which consists of a shared knowledge source which is empirically shown to improve performance in high-level coordination settings, and a *pool of policies* that agents can dynamically choose from, which helps in tackling environmental heterogeneity. We show that the proposed approach consistently outperforms established baselines MAPPO (Yu et al., 2021) and IPPO (de Witt et al., 2020) on environments across different coordination and heterogeneity levels. The knowledge source is the key to improved performance across different coordination levels whereas further ablation studies show that the pool of policies is the key to good performance across different levels of environmental heterogeneity.

## 2 RELATED WORK

**Centralized Training Decentralized Execution (CTDE).** These approaches are among the most commonly adopted variations for MARL in cooperative tasks and address the problem of environment non-stationarity in multi-agent RL. They usually involve a centralized critic which takes in global information, i.e. information from multiple agents, and decentralized policies whose learning are guided by the critic. Lowe et al. (2017) first proposed an extension of DDPG (Lillicrap et al., 2016) to a multi-agent framework by using a shared critic and agent specific policies during training, and decentralized execution. Yu et al. (2021) proposes the extension PPO (Schulman et al., 2017) to a multi-agent framework in a similar manner. Kuba et al. (2021) extends trust region learning to a cooperative MARL setting in a way that the agents do not share parameters. Foerster et al. (2018) uses the standard centralized critic decentralized actors framework with a *counterfactual baseline*. Li et al. (2021) uses an information theory-based objective to promote novel behaviors in CTDE-based approaches. Value Decomposition (Sunehag et al., 2018; Rashid et al., 2018; Wang et al., 2020; Mahajan et al., 2019), (Rashid et al., 2018) approaches learn a factorized state-action value function. Sunehag et al. (2018) proposes Value Decomposition Networks (VDN) which simply add the state-action value function of each agent to get the final state-action value function. Rashid et al. (2018) uses a mixing network to combine the action-value functions of each agent in a non-linear fashion.

**Coordination in Multi Agent Reinforcement Learning.** There have been several definitions of coordination in MARL, most of which come from the agents' perspective. Guestrin et al. (2002) and Busoniu et al. (2008) define coordination as the ability of agents to find optimal joint actions. Kapetanakis & Kudenko (2004) defines coordination as consensus among agents. Choi & Ahn (2010) defines coordination as agents' ability to achieve a common goal. In contrast, we analyze coordination levels from the angle of the RL environment. Developing MARL algorithms that train coordinated policies requires sufficient exploration due to the presence of multiple equilbria. A large section of recent approaches revolves around explicitly taking into account the states and actions of other agents by learning differentiable communication channels between agents in order to train coordinated policies. Foerster et al. (2016) proposes DRQN-based communication protocols DIAL and RIAL. Sukhbaatar et al. (2016) proposes CommNet which uses a shared recurrent module which calculates the state of each agent conditioned on the previous state and the mean of messages received from all the other agents. Jiang & Lu (2018) and Das et al. (2019) use attention based communication protocols where attention is used to transmit and integrate messages sent by other agents. Similar to our work, Kim et al. (2020a) attempts to learn coordinated behavior in a CTDE setting without introducing explicit communication, by using the mutual information of the agents' actions.

**Environmental Heterogeneity in MARL.** Environmental heterogeneity is a relatively uncharted land in MARL and has only been explored to a very limited extent in RL as a whole. Jin et al. (2022) analyzed environmental heterogeneity in a federated learning setting. The authors define heterogeneity as different state transition functions among siloed clients in the federated system, while for each client the environment is homogeneous. In another more recent study by Xie & Song (2022), heterogeneity of initial state distribution and heterogeneity of environment dynamics are both taken into consideration. Our problem is also closely related to Hidden Parameter Markov Decision Processes (HiP-MDPs) (Doshi-Velez & Konidaris, 2016) which consider a set of closely related MDPs which can be fully specified with a bounded number of latent parameters. Our approach can also be seen as being related to the multi-task reinforcement learning setting, where the goal is to learn an optimal policy that can be generalized to a set of closely related tasks. In all of the above works, heterogeneity is considered to be arising from variations across different environments or tasks. In contrast, we focus on heterogeneity within an environment.

Extended related works can be found in appendix A.1.

## 3 PRELIMINARIES

**Notation.** In this work, we consider a multi-agent version of *decentralized* Partially Observable Markov Decision Processes (Dec-POMDP) (Oliehoek & Amato, 2016). The environment is defined as $(\mathcal{N}, \mathcal{S}, \mathcal{O}, \mathbb{O}, \mathcal{A}, \mathcal{T}, \Pi, R, \gamma)$. $\mathcal{N} = \{1, ..., N\}$ denotes a set of $N > 1$ agents and $\mathcal{S}$ is the set of global states. $\mathcal{A} = A_1 \times \cdots \times A_N$ denotes the joint action space and $a_{i,t} \in A_i$ refers to the action of agent $i$ at time step $t$. $\mathbb{O} = O_1 \times \cdots \times O_N$ denotes the set of partial observations where $o_{i,t} \in O_i$ stands for

partial observation of agent $i$ at time step $t$. The joint observation $o \in \mathbb{O}$ is given by the observation function $\mathcal{O} : (a_t, s_{t+1}) \to P(o_{t+1}|a_t, s_{t+1})$ where $a_t, s_{t+1}$ and $o_{t+1}$ are the joint actions, states and observations respectively. $\Pi$ is the set of policies available to the agents. To choose actions at timestep $t$, agent $i$ uses a stochastic policy $\pi_{\theta_i}(a_{i,t}|h_{i,t})$ conditioned on its action-observation history $h_{i,t} = (o_{i,0}, a_{i,0}, \ldots, o_{i,t-1}, a_{i,t-1})$. Actions from all agents together produce the transition to the next state according to transition function $\mathcal{T} : (s_t, a_{1,t}, \ldots, a_{N,t}) \mapsto P(s_{t+1}|s_t, a_{1,t}, \ldots, a_{N,t})$. $R : \mathcal{S} \times \mathcal{A} \mapsto \mathbb{R}$ is the global reward function conditioned on the joint state and actions. At timestep $t$, the agent team receives a reward $r_t = R(s_t, a_{1,t}, \ldots, a_{N,t})$ based on the current joint state $s_t$ and the joint action $a_{1,t}, \ldots, a_{N,t}$. $\gamma$ is the discount factor for future rewards.

**Coordination level in cooperative MARL task.** Here, we quantitatively define the coordination level required in a cooperative MARL task used in this study. Recall that $R : \mathcal{S} \times \mathcal{A} \mapsto \mathbb{R}$ is the global reward function conditioned on the joint state and actions. At time step $t$, the agent team receives a reward $r_t = R(s_t, a_t)$ based on the current total state of all agents $s_t$ and joint action $a_t$. Just as in the previous section, the joint state is factorized as follows: $s_t = (s_{e,t}, s_{1,t}, \ldots, s_{N,t})$ where $s_{e,t}$ is the state of the external environment and $s_{i,t}$ is the local state of agent $i$ at timestep $t$ and $i \in \mathcal{N}$.

Let $\mathcal{G} \subset \mathcal{N}$ denote a subset of $|\mathcal{G}| = k$ agents and different subsets can overlap. We define $R_{\mathcal{G}}(s_t, a_t) = R_{\mathcal{G}}(s_{e,t}, s_t^{\mathcal{G}}, a_t^{\mathcal{G}})$ as the joint reward that can only be obtained when a subset of $k$ agents cooperate, where $s_t^{\mathcal{G}} = \{s_{i,t}\}_{i \in \mathcal{G}}$ and $a_t^{\mathcal{G}} = \{a_{i,t}\}_{i \in \mathcal{G}}$. We can then write the joint reward as the sum of rewards contributed by all subset of agents:

$$R(s_t, a_t) = \sum_{\mathcal{G} \subset \mathcal{N}} R_{\mathcal{G}}(s_{e,t}, s_t^{\mathcal{G}}, a_t^{\mathcal{G}})$$

Hence, the level of coordination $c_t$ can be defined as the positive reward that can be obtained at time $t$ if no less than $c_t$ agents are involved in it:

$$c_t = \min_{k=1,\ldots,N} \{k | \exists \mathcal{G} \subset \mathcal{N} \text{ s.t } |\mathcal{G}| = k : \quad R_{\mathcal{G}}(s_{e,t}, s_t^{\mathcal{G}}, a_t^{\mathcal{G}}) > 0\}$$

Where $|\mathcal{G}|$ is the number of elements in $\mathcal{G}$. The global coordination level of the environment $c$ can then simply be defined as: $c = \max_{t \geq 0}\{c_t\}$. This means that if there's at least one task in the environment that must be solved using the largest number of agents, then that number of agents ($c_t$) is defined as the coordination level of that environment.

It is worth mentioning the difference between the problem we explore and the formulation in previous studies such as (Zhang & Lesser, 2013). We define coordination in a way that some rewards require at least $k$ agents to coordinate with each other to obtain but different subsets of agents $\mathcal{G}_j$ do not have to be disjoint, i.e., one agent can be involved in obtaining more than one reward in a single time step.

**Heterogeneity level of cooperative MARL task.** Another aspect we like to explore is the heterogeneity of the RL environment. It is worth pointing out that the heterogeneity of the RL environment is different from the heterogeneity of agents or heterogeneity of policies as explored in previous studies (Mondal et al. (2022); Kapetanakis & Kudenko (2004).

For simplification, we define heterogeneity in a single-agent RL environment, which can be easily unwrapped into a multi-agent setting. We assume that the environment has a collection of $K$ different state-transition functions $\{\mathcal{T}_k : (s_t, a_t) \mapsto s_{t+1}\}_{1 \leq k \leq K}$. At each timestep $t$, whenever the agent takes an action, its next state is governed by one of the $K$ state-transition functions, and that choice is decided according to some (possibly latent) variable $\nu_t$. $K$ is then defined as the level of heterogeneity of the environment, if $K = 1$ then the environment is said to be homogeneous. In this study, we implement $\nu_t$ as the position of the agent in the environment, which means the state-transition function depends on where the agent is in the environment.

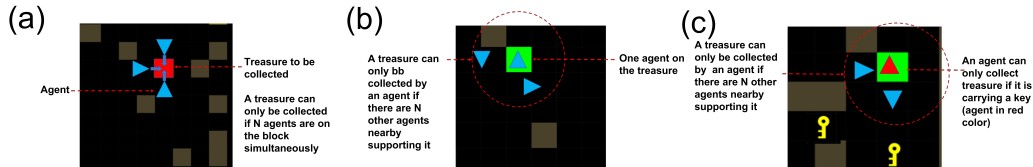

Figure 1: HECOGrid: Three cooperative multi-agent reinforcement learning environments developed in this study that allow quantitative control of coordination and environmental heterogeneity levels to adjust difficulty of cooperative tasks: (a) `TeamTogether` environment, (b) `TeamSupport` environment and (c) `KeyForTreasure` environment.

## 4  HECOGRID: MARL ENVIRONMENTS FOR VARYING COORDINATION AND ENVIRONMENTAL HETEROGENEITY LEVELS

We introduce a set of three cooperative MARL environments, which we collectively call HECOGrid, that allows manipulation of coordination and environmental heterogeneity levels in addition to several other properties. HECOGrid consists of `TeamSupport`, `TeamTogether` and `KeyForTreasure` environments as shown in Fig. 1. In this section, we describe each HECOGrid environment in detail.

**TeamTogether Environment.** The cooperative team task in this environment is to collect as many treasures as possible in a limited number of time steps. Each treasure is presented as a bright box in the environment and becomes grey once collected. In order for a treasure to be collected, a certain number of agents need to step onto it simultaneously. The number of agents required for collection is the level of coordination of the task.

**TeamSupport Environment.** This environment is similar to `TeamTogether` except that in order for an agent to collect a treasure, instead of being on the treasure together simultaneously with other agents, it needs to step onto the box and with a certain number of agents within a fixed distance (set to 2 by default) to support the collection. This number of agents required for collection support (including the agent that actually collects) is defined as the level of coordination of the task. Rewards are distributed equally across all agents in the whole team.

**KeyForTreasure Environment.** This environment is similar to `TeamSupport` except that an agent can only collect a treasure if it is carrying a key, and to collect the treasure, a certain number of agents need to be on the box simultaneously. This additional key-searching step increases the difficulty of the task. If an agent picks up a key, its color changes.

In all the environments, all the rewards are distributed equally across all agents on the team.

**Environmental heterogeneity in HECOGrid.** We implement environmental heterogeneity by dividing the grid into $K$ zones. For each zone, the transition function $\mathcal{T}$ is different. Concretely, each action leads to a different state depending on which zone the agent is in (e.g. action number 1 may make the agent turn left, right, move forward or perform some other action depending on what zone of the grid the agent is present in).

## 5  SAF: THE STATEFUL ACTIVE FACILITATOR

In this section we describe the proposed method that consists of two main components: a Knowledge Source (KS) that enhances coordination among the agents and a policy pool (PP) that allows agents to dynamically select a policy, enabling agents to exhibit diverse behaviors and have distinct goals, as well as handling heterogeneous environments.

After receiving the local observations, the agents produce messages conditioned on the observations and send them to the KS. These messages are integrated into the KS via a soft attention mechanism. This enables the KS to sift through the information shared by the agents and filter out the irrelevant information, before sending it back to the agents. The agents then utilize this message to define a more informative state which is then used by the critic. Hence, the KS acts as an information

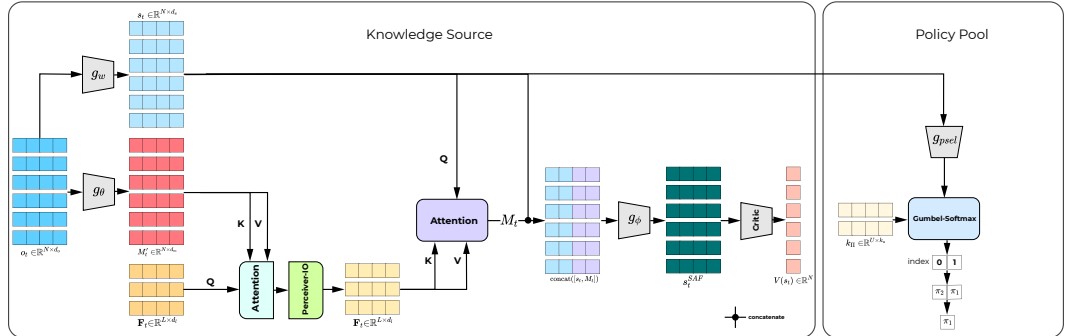

Figure 2: STATEFUL ACTIVE FACILITATOR (SAF) algorithm: during training, agents read and write from *Knowledge source (*KS*)* and pick a policy from the shared policy pool. First, each agent generates a message and competes for write-access into the KS. Next, all agents read messages from the KS and use it with their internal state as inputs into critic during training. Each agent selects their policy from a pool of policy using a trainable signature key and their states. After training, the execution is conducted in a decentralized manner as the policy is conditioned on the observation only.

bottleneck, which by filtering out irrelevant information, aids the centralized critic in coordinating the agents better. Further, each agents dynamically selects a policy from the shared pool of policies conditioned on its current state. The current state of an agent is simply an encoding of the local observation received by the agent. By using a pool of policies, we aim to train specialists which are suited to tackle different environmental conditions, hence aiding in tackling heterogeneity.

Technically, SAF can be obtained by augmenting any of the existing Centralized Training Decentralized Execution (CTDE) with the KS and PP. The setup used in our experiments closely resembles that of MAPPO (Yu et al., 2021) with the centralized critic augmented with the KS and the policies of agents replaced by a shared pool of policies. SAF is trained in the same manner as MAPPO, where the centralized critic and the KS are used for guiding the policies. These are not used during execution, which allows is to retain the CTDE nature. We train SAF end to end by using the same loss function and standard implementation practices as discussed in MAPPO in Yu et al. (2021).

In the following subsection, we explain each step. For further details, see Algorithm 1 in Appendix A.3.

**Step 1: Generating the messages.** Each agent $i$ receives a partial observation $o_{i,t}$ at each time step $t$. These observations are encoded into messages which are written into the KS by a common encoder $g_\theta$: $m'_{i,t} = g_\theta(o_{i,t})$; $m'_{i,t} \in \mathbb{R}^{d_m}$. We denote the set of messages generated by the agents at time step $t$ by $\boldsymbol{M}'_t$:

$$\boldsymbol{M}'_t = \{m'_{i,t} | 1 \leq i \leq N\}$$

**Step 2: Writing into the Knowledge Source.** The messages $\boldsymbol{M}'_t$ generated in step one are distilled into a latent state which we term as a **Knowledge Source** or KS. We represent the KS state at time step $t$ by $\boldsymbol{F}_t$. $\boldsymbol{F}_t$ consists of $L$ slots $\{l_0, l_1, ..l_{L-1}\}$, each of dimension $d_l$ so that $\boldsymbol{F}_t \in \mathbb{R}^{L \times d_l}$.

The messages in $\boldsymbol{M}'_t$ compete with each other to write into each KS's state slot via a cross-attention mechanism. The query, in this case, is a linear projection of the $\boldsymbol{F}_t$, i.e., $\widetilde{\boldsymbol{Q}} = \boldsymbol{F}_t \widetilde{\boldsymbol{W}}^q$, whereas the keys and values are linear projections of the messages $\boldsymbol{M}'_t$. KS state is updated as:

$$\boldsymbol{F}_t \leftarrow \text{softmax}\left(\frac{\widetilde{\boldsymbol{Q}}(\boldsymbol{M}'_t \widetilde{\boldsymbol{W}}^e)^{\mathrm{T}}}{\sqrt{d_e}}\right) \boldsymbol{M}'_t \widetilde{\boldsymbol{W}}^v$$

After this, self-attention is applied to the KS using a transformer encoder tower constituting a `Perceiver-IO` architecture (Jaegle et al., 2022).

**Step 3: Reading from the Knowledge Source.** The KS makes the updated state available to the agents should they deem to use it. We again utilize cross attention to perform the reading operation. All the agents create queries $\boldsymbol{Q}^s_t = \{q^s_{i,t} | 1 \leq i \leq N\} \in \mathbb{R}^{N \times d_e}$ where $q^s_{i,t} = \boldsymbol{W}^q_{\text{read}} s_{i,t}$ and

$s_{i,t} = g_\omega(o_{i,t})$ are encoded partial observations . Generated queries are matched with the keys $\boldsymbol{\kappa} = \boldsymbol{F}_t \boldsymbol{W}^e \in \mathbb{R}^{L \times d_e}$ from the updated state of SAF. As a result, the attention mechanism can be written as:

$$M_t = \text{softmax}\left(\frac{\boldsymbol{Q}_t^s \boldsymbol{\kappa}^T}{\sqrt{d_e}}\right) \boldsymbol{F}_t \boldsymbol{W}^v \tag{1}$$

where $\boldsymbol{M}_t = \{m_{i,t} | 1 \leq i \leq N\}$. Consequently, the read messages are used to define a more informative state $s_{i,t}^{\text{SAF}} = g_\phi([s_{i,t}, m_{i,t}])$, where $g_\phi$ is parameterized as a neural network. Finally, the new state $s_{i,t}^{\text{SAF}}$ is used by the critic to compute values. Interestingly, since $s_{i,t}^{\text{SAF}}$ is exclusively used by the critic, which is only used during training, SAF do not uses communication during execution.

**Step 4: Policy Selection.** In order to perform policy selection for each policy we define an associated signature key which is initialized randomly at the start of the training: $\boldsymbol{k}_\Pi = \{k_{\pi^u} | 1 \leq u \leq U\}$. These keys are matched against queries computed as deterministic function of the encoded partial observation $q_{i,t}^{\text{policy}} = g_{\text{psel}}(s_{i,t})$, where $g_{\text{psel}}$ is parametrized as a neural network.

$$\text{index}_i = \text{GumbelSoftmax}\left(\frac{q_{i,t}^{\text{policy}} (\boldsymbol{k}_\Pi)^T}{\sqrt{d_m}}\right) \tag{2}$$

As a result of this attention procedure, agent $i$ selects a policy $\pi^{\text{index}_i}$. This operation is performed independently for each agent, i.e. each agent selects a policy from the policy pool. Therefore, it does not involve communication among different agents.

## 6 EXPERIMENTS

In this section, we design empirical experiments to understand the performance of SAF and its potential limitations by exploring the following questions: (a) how much difficulty do high levels of coordination and environmental heterogeneity cause to cooperative MARL tasks? (b) does SAF perform well when coordination or/and heterogeneity levels are high? (c) Is SAF robust to changes of coordination and heterogeneity levels? and (d) SAF has two components. How does each component contribute to the performance at high coordination and heterogeneity levels?

**Baseline Methods.** We compare SAF with two widely used algorithms with related architectural designs and similar number of parameters to SAF, namely Independent PPO (IPPO) (de Witt et al., 2020) and multi-agent PPO (MAPPO) (Yu et al., 2021) (Table A.3). In IPPO, each agent has its own actor and critic and does not share information with other agents. In MAPPO, instead of being trained in a decentralized manner, the critic takes information from all agents in each step as inputs during training, and agents operate in a decentralized manner without sharing information during execution. Since, the agents in our environments are homogeneous, we use the parameter sharing for MAPPO, where the the actor and critic parameters are shared across all agents (Christianos et al., 2021; Terry et al., 2020). SAF has similar training and execution strategy as MAPPO but uses an added component - the KS before passing information to the critic during training, and a shared pool of policies instead of a single shared policy for each agent.

**High levels of coordination and environmental heterogeneity.** To understand how much difficulty high levels of coordination in the environment cause, we conducted experiments in all three HECOGrid environments for coordination levels 1 to 3 with heterogeneity level set to 1. We train all methods for 10M steps for all experiments. Our results, as shown in Figure 3 show that the performance of all three methods decreases dramatically as coordination levels increase. Performance of all three methods show a more than 50% decrease in performance at coordination level 2, as compared with coordination level 1, in all environments. At a coordination level of 3, all methods fail to show meaningful behavior. These observations indicate that tasks requiring more agents to work together for reward collection are extremely challenging in a cooperative MARL setting.

To understand how much difficulty high levels of environmental heterogeneity cause, we conducted experiments in all three HECOGrid environments for heterogeneity levels 1 to 5 with coordination level set to 1. All methods show a decrease in performance as the environments become more

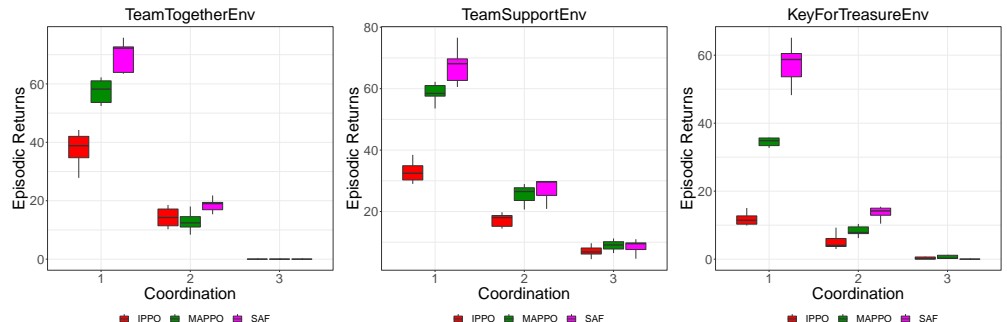

Figure 3: Test-time results for SAF, MAPPO and IPPO on `TeamTogether`, `TeamSupport` and `KeyForTreasure` environments on varying levels of coordination. The heterogeneity level is fixed at 1. Performance of all algorithms decreases as coordination levels increase with SAF showing better performance across all environments.

heterogeneous, though to a smaller extent as compared with coordination levels (see Figure 4). We provide further results for experiments performed in cases where coordination and heterogeneity levels are high simultaneously in Appendix A.2.

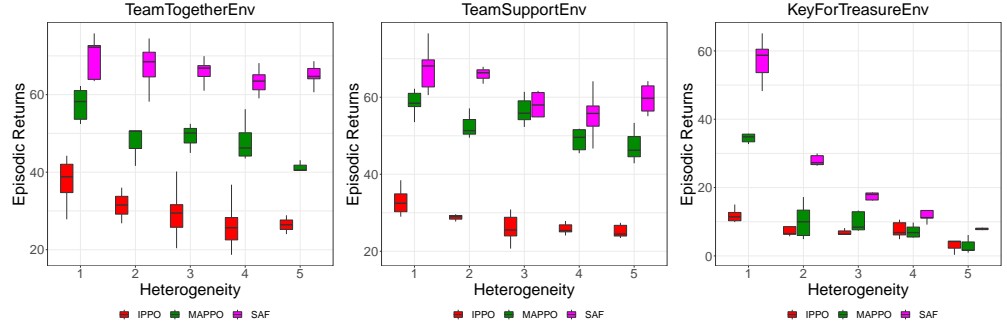

Figure 4: Test-time results for SAF, MAPPO and IPPO on `TeamTogether`, `TeamSupport` and `KeyForTreasure` environments on varying levels of heterogeneity. The coordination level is fixed at 1. All algorithms show decreased performance as heterogeneity increases. SAF shows better performance in more cases.

**Contribution of each component of SAF.** In the method section, we design SAF by hypothesizing that in order to tackle coordination and environment heterogeneity, two key elements are necessary: the use of a shared knowledge source (KS) and a shared pool of policies (PP) from which the agents can dynamically choose. We wish to understand how much each component contributes to the performance of SAF in different scenarios. To investigate this question, we conduct experiments using different ablated versions of SAF in cooperative tasks with different levels of heterogeneity and coordination. As seen in Figure 5 in the Appendix, our experimental results indicate that the knowledge source contributes to the performance of SAF in all cooperative tasks while the shared pool of policies significantly improves the performance of the agents in heterogeneous environments and has minimal contribution to tasks requiring high coordination.

## 7 CONCLUSION

In this work, we explore coordination and heterogeneity levels of cooperative MARL environments by developing a set of environments, HECOGrid, which allows full quantitative control over coordination and heterogeneity levels. Moreover, we propose a novel algorithm that enables agents to perform well in difficult environments with high levels of coordination and heterogeneity. Our experimental results suggest that high coordination and heterogeneity do make cooperative tasks challenging and our SAF method allow agents to gain better performance in these environments.

## 8 ETHIC STATEMENT AND REPRODUCIBILITY

To the best of the authors' knowledge, this study does not involve any ethical issues.The authors aim to maximize the reproducibility of the study. The codes of this project including the new environment constructed will be released in the camera-ready version. In the methods section, notions align with existing literature. A detailed description of each step in the SAF algorithm is given in the method section and a full algorithm is provided in the appendix.

## 9 ACKNOWLEDGEMENT AND AUTHOR CONTRIBUTION

Y.B. supervised the project and contributed to conceptualization, presentation and manuscript writing. N.H. contributed to the conceptualization and experimental design. M.M. contributed to conceptualization, experimental design and manuscript writing. T.S. contributed to reinforcement learning task design. A.G. contributed to project initialization, conceptualization, coordination, experimental design, method development and manuscript writing. C.M. contributed to implementation, experimental design, method development and manuscript writing. D.L. , V.S. and O.B. contributed to project initialization, conceptualization, implementation, experimental design, method development and manuscript writing.

In addition, we thank CIFAR, Mila - Quebec AI institute, University of Montreal, BITS Pilani, TUDelft, Google, Deepmind, MIT, and Compute Canada for providing all the resources to make the project possible.

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

# A APPENDIX

## A.1 ADDITIONAL RELATED WORK

**Information Bottleneck.** With the emergence of modular deep learning architectures (Vaswani et al., 2017; Goyal et al., 2021b; Scarselli et al., 2008; Bronstein et al., 2017; Kipf et al., 2018; Battaglia et al., 2018) which require communication among different model components, there has been a development of methods that introduce a bottleneck in this communication to a fixed bandwidth which helps to communicate only the relevant information. (Liu et al., 2021) use a VQ-VAE (Oord et al., 2017) to discretize the information being communicated. Inspired by the theories in cognitive neuroscience (Baars, 1988; Shanahan, 2006; Dehaene et al., 2017), (Goyal et al., 2021c) proposes the use of a generic *shared workspace* which acts as a bottleneck for communication among different components of multi-component architectures and promotes the emergence of *specialist* components. We use SAF, which is similar to the shared workspace that different agents compete to write information to and read information from.

**Communication in MARL.** Communication involves deciding which message to be shared and determining how the message-sending process is implemented. Foerster et al. (2016) and Sukhbaatar et al. (2016) implemented learnable inter-agent communication protocols. Jiang & Lu (2018) first proposed using attention for communication where attention is used for integrating the received information as well as determining when communication is needed. Das et al. (2019) uses multiple rounds of direct pairwise inter-agent communication in addition to the centralized critic where the messages sent by each agent are formed by encoding its partial observation, and the messages received by each agent are integrated into its current state by using a soft-attention mechanism. Kim et al. (2020b) uses intentions represented as encoded imagined trajectories as messages where the encoding is done via a soft-attention mechanism with the messages received by the agent. Wang et al. (2021) trains a model for each agent to infer the intentions of other agents in a supervised manner, where the communicated message denotes the intentions of each agent. The above-mentioned approaches require a computational complexity that is quadratic in the number of agents whereas our approach has a computational complexity that is linear in the number of agents. Moreover, we show that our approach is able to outperform several standard baselines using messages which can be computed as simply encoding each agent's partial observation. Weis et al. (2020) developed a transformer-based multi-agent reinforcement learning method that models MARL decision-making as a sequential model.

**Coordination in MARL and the Pareto-optimal Nash equilibrium**. In the field of MARL, coordination is usually defined as the ability for agents to make optimal decisions to achieve a common goal by finding an optimal joint action in a dynamic environmentChoi & Ahn (2010); Kapetanakis & Kudenko (2004). One way to find the optimal joint actions by a group of agents is by studying the Pareto-optimal Nash equilibrium Zhukovskiy & Kudryavtsev (2016), which describes the optimal solution as one in which no agent's expected gain can be further increased without compromising other agents' gain. However, there exist several challenges in cooperative MARL systems to achieve Pareto-optimal solutions. In the following sections, we are going to explain three of these challenges, which are the ones we seek to tackle in this study as well as their links to coordination and environmental heterogeneity levels.

**Environmental heterogeneity and the non-stationarity problem** In MARL, the transition probabilities associated with the action of a single agent change over time as the action dynamics of the other agents change (Bowling & Veloso, 2000). To solve this problem of non-stationarity, most recent MARL methods follow the Centralized Training Decentralized Execution paradigm. The most extreme case of centralized training is when all agents share the same set of parameters. However, parameter sharing also assumes that all agents have the same behaviors, which is not true when there is heterogeneity either among the agents themselves or in the environment. Previous studies use indicator-based methods to personalize a shared policy in a group of heterogeneous agents(Terry et al., 2020), however, environmental heterogeneity has been less explored in the literature (Jin et al., 2022).

**Environmental heterogeneity and the alter-exploration problem** Another problem environmental heterogeneity may cause in cooperative MARL is the alter-exploration problem. The balance between exploration and exploitation is crucial for all reinforcement learning tasks. In cooperative MARL this problem arises when exploration of one agent may penalize other agents and their corresponding

policies during training as the cooperative agents share rewardsBoutoustous et al. (2010); Matignon et al. (2012). Environmental heterogeneity could potentially lead to worse alter-exploration problems as there tend to be more unseen states for an exploring agent which may result in higher and more frequent penalties. In this study, we seek to solve the above-mentioned problem using a combination of inter-agent communication via an active facilitator and a shared pool of policies.

**Existing Environments in MARL.** Some of the existing benchmarks based on online multiplayer games, attempt to move away from the toy-like grid world setting for MARL environments in favor of more realistic environments, by making use of high-dimensional observation and action spaces, continuous action spaces, challenging dynamics, and partial observability (Vinyals et al., 2017; Berner et al., 2019; Leibo et al., 2021; Zheng et al., 2017). These benchmarks focus on decentralized control in cooperative tasks and agents are heterogeneous (i.e., different types of agents having different abilities). In principle, it is possible to vary the levels of coordination and heterogeneity since the difficulties of different environments vary. However, there is no well-defined notion of the two concepts and these can't be varied in a controlled fashion. MeltingPot, which was recently proposed in (Leibo et al., 2021) focuses on test-time generalization abilities of a group of agents includes a wide range of scenarios: competitive games, games of pure common interest, team-based competitive games, and mixed motion games which stress test the coordination abilities of the agents. However, similar to other benchmarks, there is no systematic decomposition nor a quantitative notion of the concepts of *coordination* and *environmental heterogeneity*.

## A.2   ADDITIONAL RESULTS

In this section, we show the training curves for SAF, MAPPO and IPPO on `KeyForTreasure`, `TeamSupport` and `TeamTogether` environments. We additionally present more ablation results in the Out-of-Distribution setting.

**Performance when increasing coordination or/and heterogeneity levels.** SAF shows significant performance improvement upon MAPPO and IPPO at coordination levels 1 and 2 (Figure 3). In addition, SAF shows faster performance increase at an early stage of the training process (figure A.2). This suggests potential advantages for training agents using SAF in cooperative tasks requiring high coordination. At most heterogeneity levels, SAF shows performance improvement upon MAPPO and IPPO in all HECOGrid environments and a faster increase in performance at early stages of the training (Figure 4 and A.2). This suggests potential advantages for training agents using SAF in a cooperative environment which are heterogeneous. In addition to manipulating coordination and heterogeneity levels separately, experiments are conducted to understand if SAF can perform well in environments in which both parameters are high. In the relatively easy `TeamSupport` environment with both coordination and heterogeneity set at 2, 3 and 4, SAF again shows improved performance over IPPO and MAPPO (Figure 6(c)). Figure 6(a) shows the training curves for SAF, MAPPO, and IPPO on `KeyForTreasure`, `TeamSupport` and `TeamTogether` environments for a coordination level of 2. It shows a gap in performance between SAF and the baselines and this gap is further enlarged when it comes to a heterogeneity level of 2 (see Figure 6(b)) which shows that SAF is effectively able to handle changes in the environment's dynamics.

**Robustness to changes in coordination and heterogeneity levels.** In most real-world applications, such as robots in warehouses, the coordination levels as well as environmental heterogeneity levels can change over time and may even be unknown to the agents. Therefore, the agents' robustness to such changes is important. To understand if agents trained with SAF can still function well in these out-of-distribution (OOD) settings, we conduct experiments to test the agents' performance on `TeamSupport` and `TeamTogether` environments with heterogeneity or coordination levels that are different as compared to the ones used during training. First, we train the agents in environments with a coordination level of 2 and a heterogeneity level of 1 and test their performance at coordination levels between 1 and 3, and a heterogeneity level of 1. As shown in Figure 7, SAF shows better transfer in the more difficult `TeamTogether` environment than other methods but fails to perform as well as MAPPO in the `TeamSupport` environment. Next, we train the agents in environments with a coordination level of 1 and a heterogeneity level of 2 and test their performance at a coordination level of 1 and heterogeneity levels between 1 and 5. As shown in figure 7, SAF shows superior performance in `TeamTogether` environment and matches the performance of MAPPO in `TeamSupport` environments. This suggests that SAF has similar robustness to changes in coordination and heterogeneity levels as some of the widely used baselines in the MARL community.

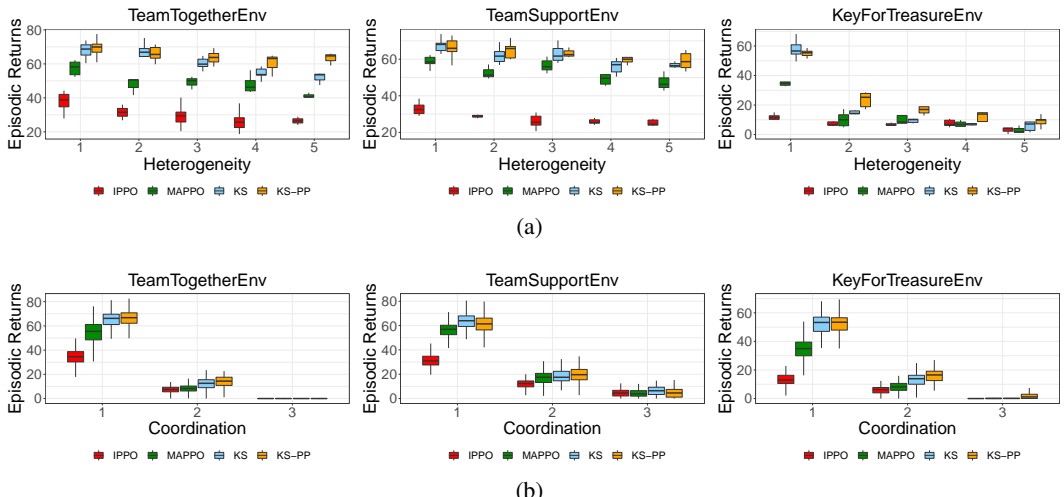

(a)

(b)

Figure 5: Ablation study to understand the contribution of the Knowledge Source (KS) and the shared policy pool (PP) to the performance of SAF in HECOGrid environments. In the legend, KS indicatess SAF without the Pool of Policies whereas KS-PP essentially means SAF (a) Performance of the KS and the pool of policies against baselines in increasing levels of heterogeneity. (b) Performance of the KS and the pool of policies against baselines in increasing levels of coordination. KS contributes to performance in all settings and PP especially improves performance in heterogeneous environments.

**Ablation Study for OOD generalization**    Figure 8(a) shows test-time generalization results on the `TeamTogether` and `TeamSupport` environments where the training coordination level was set to 2 and the heterogeneity was set to 1. The pool of policies in SAF is important in getting good performance especially when it's tested on levels of coordination not seen during training. Moreover, Figure 8(b) further validates that the pool of policies is important in handling varying environment dynamics as SAF was trained on a heterogeneity level of 2 and a coordination level of 1 and the results on unseen levels of heterogeneity are better than SAF trained without a pool of policies. These ablations show that the introduced pool of policies in SAF is key to its performance.

**Comparison with QPLEX baseline** Performance of QPLEX and QPLEX with a shared pool of policies are compared in different environments with 5 agents. The reason only pool of policies but not share knowledge source was used is because QPLEX already has a similar mechanism. The results suggest that a shared pool of policies among agents improve learning efficiency when coordination level is high (Figure 9).

## A.3    ALGORITHM

In this section, we summarize the components of SAF  and its computations in the form of a pseudo-code outlined in algorithm 1.

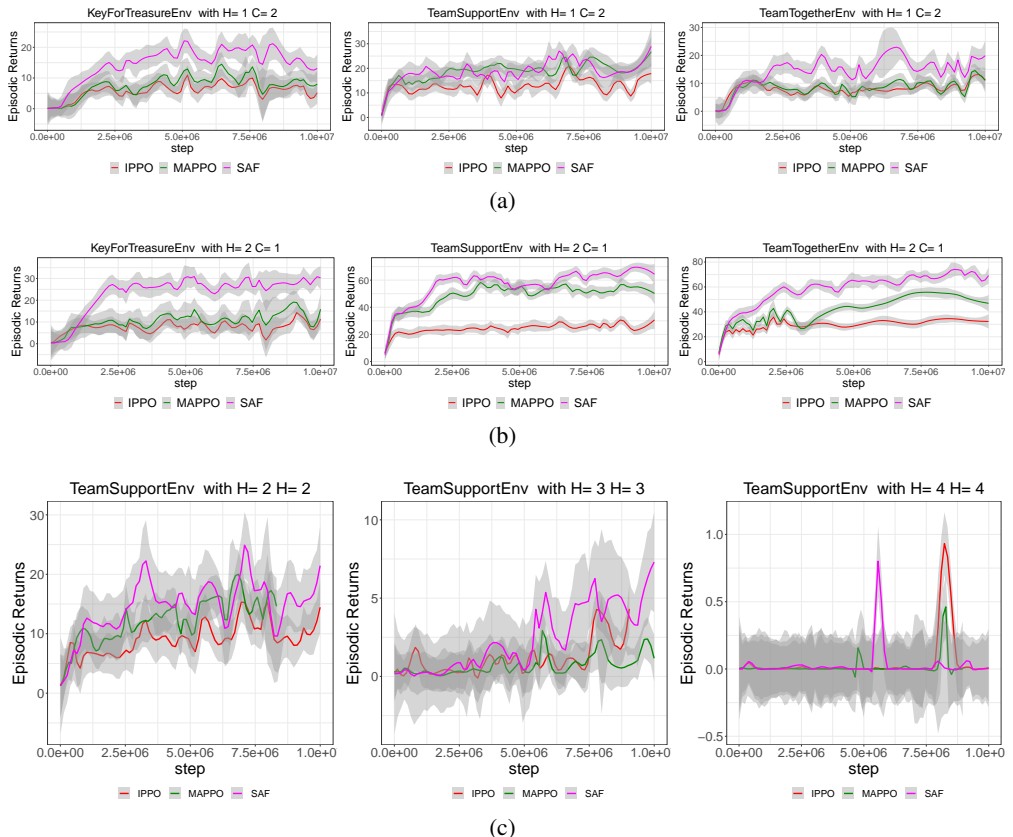

Figure 6: Examples of training curves for SAF, MAPPO and IPPO on the `KeyForTreasure`, `TeamSupport` and `TeamTogether` at different coordination and Heterogeneity levels. At the initial stage of the SAF show a faster increase in performance. After convergence, SAF shows improved performance in most tasks compared to IPPO and MAPPO.

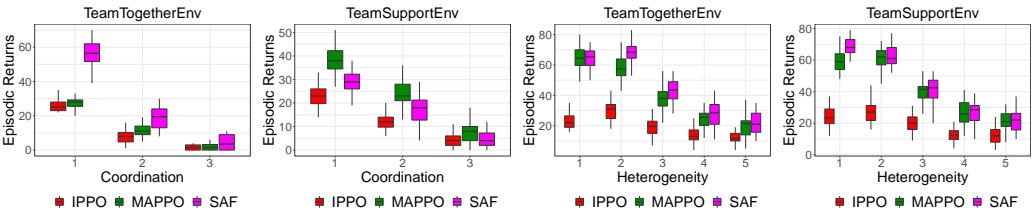

Figure 7: Out-of-Distribution generalization study to understand the robustness of SAF to changes in coordination and heterogeneity levels in HECOGrid environments. Agents are trained at certain heterogeneity and coordination levels, and tested on unseen levels. In general, SAF matches MAPPO in robustness to shifts in coordination or heterogeneity level

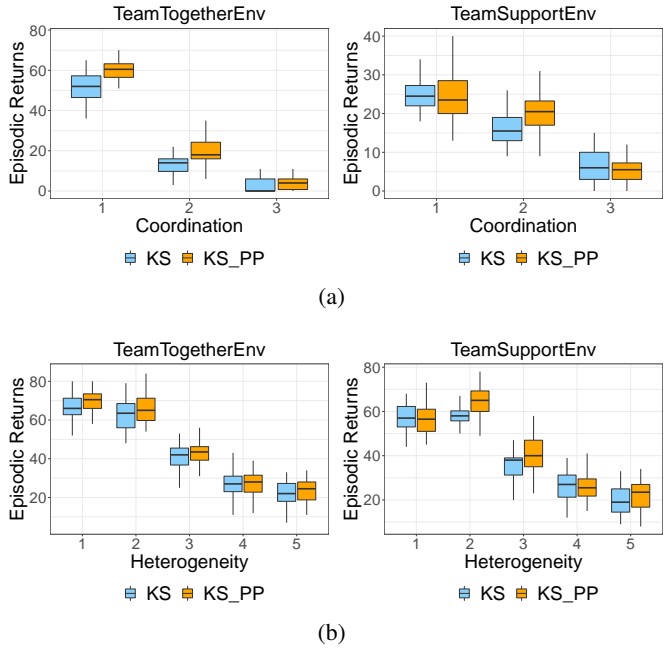

(a)

(b)

Figure 8: Ablation study for Out-of-Distribution generalization. In order to understand the robustness of different ablated models to shifts in either coordination or heterogeneity levels, models trained with coordination level 2 and heterogeneity level 1 are tested across different coordination levels (Figure 8(a)). In a similar manner, The models trained on coordination level 1 and heterogeneity level 2 are tested across different heterogeneity levels (Figure 8(b)). KS indicates SAF without pool of policies whereas, KS_PP means SAF.

## A.4 IMPLEMENTATION DETAILS

In this section, we present the necessary implementation details for reproducing our results. We first present the algorithm's hyperparameters, next, we present the architectures used for each algorithm, finally, we present the environments hyperparameters.

### A.4.1 ALGORITHMS HYPERPARAMETERS

In this section, we present the relevant hyperparameters related to training the algorithms showcased in our paper. The hyperparameters shown in this section are kept fixed throughout all three environments. MAPPO, IPPO and SAF are all trained using Proximal Policy Optimization and Table A.4.1 summarizes the training hyperparameters for each algorithm. IPPO and MAPPO are trained with Generalized Advantage Estimation (GAE) while SAF is not. All algorithms are trained with Adam optimizer with a fixed learning rate of $0.0007$ throughout training, a weight decay of 0 and $\epsilon = 10^{-5}$.

### A.4.2 ARCHITECTURAL HYPERPARAMETERS

In this section, we present the exact architectures along with the hyperparameters that were used for each algorithm.

**Actor and Critic Architectures**    Listing 1 illustrates the pytorch-style implementations of the actor and critic architectures. While MAPPO shares parameters for both the actor and the critic, IPPO trains separate networks for both the actor and the critic. SAF uses the same architecture (and the same hyperparameters) as IPPO and MAPPO for the actor network, with the difference that SAF initializes a pool of policies for each agent. Listing 2 shows a pytorch-style implementation of a CNN that acts as our feature extractor. For IPPO and MAPPO, each agent's observation is fed to the CNN to generate a feature vector $z \in \mathbb{R}^{N \times C}$ where $N = 10$ is the number of agents and $C = 64$

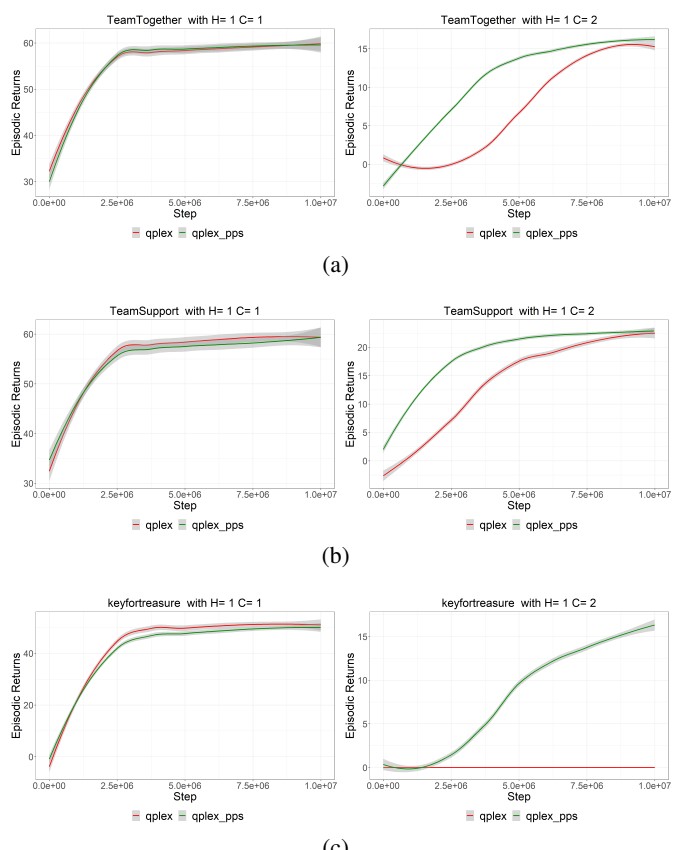

Figure 9: Performance comparison between QPLEX baseline and QPLEX with pool of policies in different environments with 5 agents. The reason only pool of policies but not share knowledge source was used is because QPLEX already has a similar mechanism. The results suggest that a shared pool of policies among agents improve learning efficiency when coordination level is high

**for** $t \leftarrow 1$ **to** $T$ **do**

    **Step 1: Each agent $i$ having state information $o_{i,t}$ (encoded partial observation), generates a message**.

    $\forall i \in \{1, \ldots, N\}, m'_{i,t} = g_\theta(o_{i,t})$
    $\boldsymbol{M}'_t = (m'_{1,t}, m'_{1,t}, m'_{2,t} \ldots, m'_{N,t})$

    **Step 2: KS integrates information from all agents**

    $\widetilde{\boldsymbol{Q}} = \boldsymbol{F}_t \widetilde{\boldsymbol{W}}^q$
    $\boldsymbol{F}_t \leftarrow \text{softmax}\left( \frac{\widetilde{\boldsymbol{Q}}(\boldsymbol{M}'_t \widetilde{\boldsymbol{W}}^e)^{\mathrm{T}}}{\sqrt{d_e}} \right) \boldsymbol{M}'_t \widetilde{\boldsymbol{W}}^v$
    KS state $F_t$ is then updated trough self-attention

    **Step 3: Information from KS is made available to each agent**

    $q^s_{i,t} = \boldsymbol{W}^q_{\text{read}} s_{i,t}, \forall i \in \{1, \ldots, N\}$
    $\boldsymbol{\kappa} = (\boldsymbol{F}_t \boldsymbol{W}^e)^T$
    $\boldsymbol{M}_t = \text{softmax}\left( \frac{\boldsymbol{Q}^s_t \boldsymbol{\kappa}}{\sqrt{d_e}} \right) \boldsymbol{F}_t \boldsymbol{W}^v$

    **Step 4: Policy Selection from the pool**

$$\forall i \in \{1, \ldots, N\} \begin{cases} q^{policy}_{i,t} & = g_{psel}(s_{i,t}) \\ \text{index}_i & = \text{GumbelSoftmax}\left( \frac{q^{policy}_{i,t}(\boldsymbol{K}^{\Pi}_t)^T}{\sqrt{d_m}} \right) \\ a_{i,t} & = a_{i,\text{index}_i,t} \end{cases}$$

**end**

**Algorithm 1:** Detailed algorithm for learning MARL policies with SAF

| Method | No. of parameters |
|---|---|
| MAPPO | 2,477,296 |
| IPPO | 1,740,016 |
| SAF (Our Method) | 2,698,342 |

Table 2: Comparing the number of parameters used in our implementations of the discussed approaches

| | SAF | MAPPO | IPPO |
|---|---|---|---|
| Learning rate | 0.0007 | 0.0007 | 0.0007 |
| PPO update epochs | 10 | 10 | 10 |
| Number of minibatches | 1 | 1 | 1 |
| Discount rate $\gamma$ | 0.99 | 0.99 | 0.99 |
| GAE | No | Yes | Yes |
| GAE's $\lambda$ | - | 0.95 | 0.95 |
| Entropy loss coefficient | 0.01 | 0.01 | 0.01 |
| Value loss coefficient | 0.5 | 0.5 | 0.5 |
| Advantage Normalization | Yes | Yes | Yes |
| Value loss clipping value | 0.2 | 0.2 | 0.2 |
| Gradient norm clipping value | 9 | 10 | 10 |
| Value loss coefficient | 0.5 | 0.5 | 0.5 |
| Optimizer | Adam | Adam | Adam |
| Optimizer's epsilon ($\epsilon$) | 1e-5 | 1e-5 | 1e-5 |
| Weight decay | 0 | 0 | 0 |

Table 3: Hyperparameters used for training the MARL algorithms across all the HECOGrid environments.

the hidden dimension. The feature vector $z$ is fed to the actor network to get the action probabilities. For IPPO, the feature vector $z$ is fed as is to the critic to get the value function, while for MAPPO, a vector $\tilde{z} = \texttt{concatenate}(z, \texttt{dim}=-1) \in \mathbb{R}^{NC}$ is formed by concatenating feature vectors from all agents and then fed to the critic, that's why we make the distinction between different critic architectures in Listing 1.

```python
from torch import nn

n_agents = 10

def layer_init(layer, std=np.sqrt(2), bias_const=0.0):
    nn.init.orthogonal_(layer.weight, std)
    nn.init.constant_(layer.bias, bias_const)
    return layer
# actor's output is a vector of 7 channels which corresponds to the
    number of actions.
actor = nn.Sequential(
    layer_init(nn.Linear(64, 128)),
    nn.Tanh(),
    layer_init(nn.Linear(128, 128)),
    nn.Tanh(),
    layer_init(nn.Linear(128, 7))
)

critic_ippo = nn.Sequential(
    layer_init(nn.Linear(64, 128)),
    nn.Tanh(),
    layer_init(nn.Linear(128, 128)),
    nn.Tanh(),
    layer_init(nn.Linear(128, 1))
)

critic_mappo = nn.Sequential(
    layer_init(nn.Linear(64 * n_agents, 128)),
    nn.Tanh(),
    layer_init(nn.Linear(128, 128)),
    nn.Tanh(),
    layer_init(nn.Linear(128, 1))
)
```

Listing 1: Pytorch-style implementation of the actor and critic architectures with the hyperparameters used in the paper for IPPO and MAPPO. We also provide implementation for the orthogonal initialization scheme.

```python
from torch import nn
import torch.nn.functional as F

class CNN(nn.Module):
    def __init__(
        self,
        in_channels=3,
        channels=[32, 64],
        kernel_sizes=[4, 3],
        strides=[2, 2],
        hidden_layer=512,
        out_size=64):

        super().__init__()

        self.conv1 = nn.Conv2d(in_channels, channels[0], kernel_sizes[0],
    strides[0])
        self.conv2 = nn.Conv2d(channels[0], channels[1], kernel_sizes[1],
    strides[1])
        self.linear1 = nn.Linear(2304, hidden_layer)
```

```
19          self.linear2 = nn.Linear(hidden_layer, out_size)
20
21      def forward(self, inputs):
22          x = F.relu(self.conv1(inputs / 255.))
23          x = F.relu(self.conv2(x))
24          x = x.reshape(x.shape[0], -1)
25          x = F.relu(self.linear1(x))
26          x = self.linear2(x)
27
28          return x
```

Listing 2: Pytorch-style implementation of the CNN that generates a feature vector from the observations. The feature vector is then input to the actor and the critic.

**Mapping function Architectures** SAF makes use of MLPs architectures to encode the partial observations, messages, policies and to combine encoded state and encoded messages. Each agent's observation is encoded with a $g_\theta$ that is a CNN (see Listing 2) which generates a feature vector $z$. The agent's states are derived from the observation encoding using $g_w$ which is the State Projector. $g_\phi$ projects the concatenation of the agent's state and message. Finally, $g_{psel}$ is an MLP that encodes $z$ and is used to select a policy from the pool of policies. See Table A.4.2 for an overview of the MLP architectures and their hyperparameters.

| $g_\theta$ | $g_w$ | $g_{psel}$ | $g_\phi$ |
|---|---|---|---|
| FC(64) | FC(64) | FC(64) | FC(64) |
| FC(128) | FC(128) | FC(128) | FC(128) |
| Input State [64] | Input State [64] | Input State [64] | Input State [128] |

Table 4: Mapping functions used to encode partial observations, messages, policies and to combine encoded state and encoded messages.

**Knowledge Source Hyperparameters** We present the hyperparameters used in the `Perceiver-IO` architecture (Jaegle et al., 2022) that makes up the knowledge source. We use 2 Perceiver layers and use $L = 4$ slots for the knowledge source. The number of policies in the pool is set to $4$. Table A.4.2 summarizes the hyperparameters used to define both the Perceiver Encoder and the Cross Attention Layer.

| `Perceiver-IO` **Hyperparameters** | Values |
|---|---|
| **PerceiverEncoder Hyperparameters** | |
| latent dimension | 4 |
| num latent channels | 64 |
| cross attention channels | 64 |
| self attention heads | 1 |
| self attention layers per block | 1 |
| self attention blocks | 1 |
| dropout | No |
| **CrossAttention Hyperparameters** | |
| query dimension | 64 |
| key, value dimension | 64 |
| num query, key channels | 64 |
| num value channels | 64 |
| dropout | No |

Table 5: Hyperparameters used to define the `Perceiver-IO` architecture used within the Knowledge Source.

### A.4.3 ENVIRONMENT HYPERPARAMETERS

This section presents the hyperparameters for our three environments excluding coordination/heterogeneity levels since those are experiment dependent and are clarified in the main text. The agents

| | TeamTogether | TeamSupport | KeyForTreasure |
|---|---|---|---|
| Gym Observation Space | | Box(0,255,(28,28,3),dtype=uint8) | |
| Gym Action Space | | Discrete(7) | |
| Number of treasures | 100 | 100 | 100 |
| Grid size | 30 | 30 | 30 |
| Max. number of steps/episode | 50 | 50 | 50 |
| Partial View Size | 7 | 7 | 7 |
| View Tile Size | 4 | 4 | 4 |
| Clutter Density | 0.1 | 0.1 | 0.1 |

Hyperparameters used in the paper for all three environments. The partial view size parameter controls how much of the grid the agent can see.

