# OpenReview forum: "Stateful Active Facilitator: Coordination and Environmental Heterogeneity in Cooperative Multi-Agent Reinforcement Learning"
_ICLR.cc/2023/Conference — ICLR 2023 poster_

### Official Review · Reviewer_y8q8 · 2022-10-25

**Confidence:** 4
**Correctness:** 3
**Technical Novelty And Significance:** 3
**Empirical Novelty And Significance:** 3
**Recommendation:** 6

**Clarity, Quality, Novelty And Reproducibility:**

The paper seems to be attempting to do too much in a single paper. The authors are commended for developing a set of three new environments to test the efficacy of MARL algorithms along dimensions of heterogeneity and cooperation in terms of the environment. The authors are also commended for additionally developing a new MARL approach. However, doing both in one paper means that a significant amount of detail is left to appendices/supplementary detail. As such, it is difficult to fully understand the details of the technical approach. I would recommend either figuring out a way to move some of the ablation studies (and/or Figure 2) to the appendix and instead moving more details about the algorithm and its motivation/intuition and psuedocode to the main paper.

The novelty is clear in terms of a new set of domains, but the novelty w.r.t. MARL algorithms is unclear without further empirical analysis against more baselines (or better discussion of why those baslines are not necessary).

Code for reproducibility appears missing.

**Strength And Weaknesses:**

Strengths:
-This paper clearly states its contributions on Page 3 (and in the abstract).
-Unlike many papers in MARL, this paper contributes not only an algorithm but also a set of MARL environments that push MARL algorithms' ability to handle environment heterogeneity and test the need of these algorithms to truly coordinate.
-The domains presented (TeamTgoether, TeamSupport and KeyForTreasure) are intuitive and do, indeed, appear to require coordination (unlike some MARL domains). Though, it is important to note that these are not the only domains that *benefit* from coordination (e.g., Predator-Prey [1]).
-The paper shows positive results for SAF vs. MAPPO and IPPO on the proposed benchmark environments while varying the degree of coordination (Figure 3) and heterogeneity (Figure 4).
-The paper also provides generally positive (but somewhat mixed results) for out-of-distribution generalization (Figure 5).
-The breadth and depth of the empirical results in this paper (not including the appendices) is impressive.

Weaknesses:
-The paper could be improved with a DEC-POMDP setup formalism to help explain the terminology/line-of-inquiry.
-The description of the Stateful Active Facilitator (SAF) is relatively short. While more details are in the appendix, it would be helpful if pseudocode was provided (Algorithm 1) in the main paper. Figure 2 does not help much in describing the approach and could be removed to make room for a better illustration or more text-based details. Many approaches work by generating and integrating messages -- how does this approach really differ? The technical uniqueness is unclear here in the algorithmic formulation. The selection of predefined(?) policies appears unique, but details are missing about how these policies are enumerated and trained.
-The paper is missing important baselines in the MARL literature, including [2], [3], [4], and [5]

[1] Singh, A., Jain, T. and Sukhbaatar, S., 2018. Learning when to communicate at scale in multiagent cooperative and competitive tasks. arXiv preprint arXiv:1812.09755.

[2] Rashid, T., Samvelyan, M., Schroeder, C., Farquhar, G., Foerster, J. and Whiteson, S., 2018, July. Qmix: Monotonic value function factorisation for deep multi-agent reinforcement learning. In International conference on machine learning (pp. 4295-4304). PMLR.

[3] Seraj, E., Wang, Z., Paleja, R., Martin, D., Sklar, M., Patel, A. and Gombolay, M., 2022, May. Learning efficient diverse communication for cooperative heterogeneous teaming. In Proceedings of the 21st International Conference on Autonomous Agents and Multiagent Systems (pp. 1173-1182).

[4] Iqbal, S. and Sha, F., 2019, May. Actor-attention-critic for multi-agent reinforcement learning. In International conference on machine learning (pp. 2961-2970). PMLR.

[5] Mao, W., Yang, L., Zhang, K. and Basar, T., 2022, June. On improving model-free algorithms for decentralized multi-agent reinforcement learning. In International Conference on Machine Learning (pp. 15007-15049). PMLR.

**Summary Of The Paper:**

This paper addresses the problem of Multi-agent Reinforcement Learning (MARL). This paper specifically addresses two components of MARL. First, the paper designs and presents three benchmark domains to explore the limits of modern MARL algorithms in terms of their ability to handle environment heterogeneity and the need to actually cooperate (rather than independently acting to complete the team's task). Second, the paper develops a new algorithm for MARL specifically designed for heterogeneity and cooperation. The paper evaluates their approach against baseline methods in the new benchmark domains, presenting numerous, positive results.

**Summary Of The Review:**

This paper contributes three new domains to explore the limits of MARL in terms of heterogeneity and cooperation. These domains will likely be high-impact for the community. The paper also contributes an algorithm. The algorithm is under-described, unfortunately. Either, the paper must make more room for details about the technical algorithm (and adding baselines to demonstrate the technical contribution), or the paper needs to be refocused on evaluating MARL algorithms on these domains to show which methods are capable of heterogeneity/cooperation and why/why not.

---

### Official Review · Reviewer_PAoT · 2022-10-25

**Confidence:** 3
**Correctness:** 3
**Technical Novelty And Significance:** 3
**Empirical Novelty And Significance:** 3
**Recommendation:** 6

**Clarity, Quality, Novelty And Reproducibility:**

*Clarity: The paper is overall easy to follow
*Novolty: This work is somewhat novel, but should be better motivated
*Reproducibiliy: Code not provided so this is unknown.

**Strength And Weaknesses:**

**Strength**
- This work tries to study and define the level of coordination in MARL, which can be a good perspective to evaluate various algorithms.
- The paper is overall easy to follow.

**Weakness**
- For the definition of the coordination level, it seems each time step is considered in isolation, what about coordinations that are temporal, such as the actions that need to be taken by one agent after another agent has taken a certain action?
- A high-level diagram should be provided for the proposed algorithm which would make it simpler for the readers to understand the overall network architecture.
- It would a good idea to further motivate how heterogeneous environments can happen in a multi-agent environment. Why would different agents have different transition functions (designed as the change of action order)?  How does that map to a real-world environment?
- The topic of the work is related to coordination and hetergeneous environment, but it is not so well motivated why these two are explored simultaneously in a single paper. The designed environments seem to focus more on the coordination level while the designed algorithm (dynamically selects model) seems to focus on the hetergeneous property.
- From the evaluation result, it seems the designed solution performs better for different hetergeneous environments, but has less improvement when the coordination level is higher. How does the proposed algorithm improve coordination?


**Summary Of The Paper:**

This explores training MARL under heterogeneous environments with different coordination difficulties. The authors first define heterogeneous environments and different levels of coordination. Then three Grid environments are designed to support different levels of coordination. Then the author proposed an attention-based method to solve the designed tasks.

**Summary Of The Review:**

The work investigates the training of MARL under different heterogeneous environments and coordination levels. The authors have a good attempt to define the coordination level of MARL, though it seems incomplete. The relationship between the environment heterogeneity and coordination and motivation should be discussed more in the paper.

---

> ### Author Response · Authors · 2022-12-12
> **Updated Impression ?**
>
> Dear. Reviewer,
>
> Thank you for your help and time during the rebuttal process.
>
> We did more experiments (as other reviewers requested). We would like to share our new results on the value based QPLEX method, which is known to be a very strong baseline. Our method QPLEX-PPS ( with the pool of policy) show higher training efficiency when coordination is high. Please see the anonymous file sharing link here for plots:https://anonymous.4open.science/r/Newresults-B9CD/QPLEX_marlgrid_line_n_agents5_env_TeamSupport3_1.png. [Reviewer UJ6m have increased their score in lieu of the recent updated results].
>
> Since the discussion time is about to end, we are wondering if we have addressed your concerns ? We are also very happy to provide any further information.
>
> Thank you for your time, and help.

---

> ### Comment · Reviewer_PAoT · 2022-12-12
> **Reply to Authors**
>
> I would like to thank the authors for the reply and revision. I think the paper is better with the revision and additional experiments. I have updated my score.

---

### Official Review · Reviewer_UJ6m · 2022-10-31

**Confidence:** 4
**Correctness:** 3
**Technical Novelty And Significance:** 2
**Empirical Novelty And Significance:** 3
**Recommendation:** 6

**Clarity, Quality, Novelty And Reproducibility:**

**Reproducibility**: The reproducibility of the paper is very poor. Very little information about the algorithm is included. The source code is not included. The expectation is also that when the authors propose a benchmark, they would include the source code.

**Clarity**: Most of the paper is clearly written (in terms of the use of language). However, it lacks organization in several places.
Several details are missing about SAF. I would also recommend adding information on how to use the new environment suite in the appendix.

**Quality**: While the contributions of the paper are worthwhile the quality of the work cannot be fully assessed with the lack of reproducibility.

**Novelty**: HECOGrid is a very welcoming contribution since it is currently missing from the literature an environment with these properties. SAF seems like a standard CTDE algorithm with an extra attention module and some additional implementation tricks. It is not clearly explained in the text, why we expect SAF to perform better compared to other baselines.

**Strength And Weaknesses:**

Strengths

- The paper proposes an interesting suite of environments that would be useful to the MARL community.
- The evaluation of SAF is detailed and the evaluation baselines are appropriate.

-------------------------------------------------------------------------------
Weaknesses
- In Table 1, MPE has scenarios with partial observability, e.g. simple_speaker_listener
- Page 3, paragraph Notation: Why are the policies of the agents conditioned on the observation and not the history of observations?
- SAF is not properly described in the text. What is the actor's loss? Is it trained using the MAPPO loss? Is it an on-policy or an off-policy algorithm?
- The used hyperparameters are not mentioned anywhere in the paper.
- How does SAF compare to SOTA Value Decomposition methods, for example, QPLEX?
- Do you use parameter sharing for the algorithms? If not, why? It has been shown in several studies that parameter sharing significantly improves the achieved returns of cooperative MARL with homogeneous agents.
- The paper claims that one of its main contributions is HECOGrid, a suite of MARL environments. However, the authors have not included the source code of HECOGrid.
- While state/observation heterogeneity is a feature currently missing from many MARL environments, I believe that agent heterogeneity is a more interesting and fundamental challenge (for the MARL community).
- It seems like some IPPO results are missing from Figure 4.
- Most of the experiments are performed for coordination level 1, which means that only a single agent is required to solve the task. That primarily shows that SAF does not "enables agents to work efficiently in high-coordination and high-heterogeneity environments". It seems that the main benefits of SAF come from the centralized computation of the value function. Can the authors elaborate more on why they believe that SAF outperforms the related baselines?
- In Figure 6, what do KS, and KS-PP mean? Does KS mean SAF without PP? What is KS-PP? Is it just SAF? Or is it SAF without KS? If this is the case, how much is the contribution of KS to SAF?
- I would expect to see more clarifications on why SAF is better compared to baselines. This is not presented in the paper. I can only see the algorithm description without much justification behind the design choices.



**Summary Of The Paper:**

The authors propose HECOGrid, a suite of MARL environments. The authors claim that two properties that affect the performance of MARL algorithms in the environment are coordination and heterogeneity. The authors formally define these two metrics. HECOGrid allows the user to choose the level of coordination and heterogeneity in the environment. The authors also propose Stateful Active Facilitator (SAF), a cooperative MARL algorithm. They evaluate SAF in HECOGrid, for varying levels of coordination and heterogeneity against IPPO and MAPPO.

**Summary Of The Review:**

Overall, I believe that the paper presents some worthy contributions. However, I have several concerns about the reproducibility of this work. Additionally, the writing seems a bit rushed, since the paper is not well organized. Additionally, the lack of some results in the figures, as well as, the lack of the source code shows that this work is in its early stage. Therefore, I recommend rejection. I would happily increase my score if the authors address my concerns.

---

> ### Author Response · Authors · 2022-12-12
> **improvement upon QPLEX in all 3 envs**
>
> Dear reviewer
> We have finished experiments on applying our methods on QPLEX on all three envs. The results show improvement upon QPLEX in all three envs when coordination level is high
> Please see the results here:
> https://anonymous.4open.science/r/Newresults-B9CD/QPLEX_marlgrid_line_n_agents5_env_TeamSupport3_1.png
>
> Minimum differences are seen at high level of heterogeneity

---

> ### Comment · Reviewer_UJ6m · 2022-12-12
> **Response to Authors**
>
> I would like to thank the authors for improving the paper. I have increased my score to 6.

---

### Official Review · Reviewer_RJCt · 2022-11-03

**Confidence:** 4
**Correctness:** 3
**Technical Novelty And Significance:** 2
**Empirical Novelty And Significance:** 3
**Recommendation:** 8

**Clarity, Quality, Novelty And Reproducibility:**

The paper is written reasonably clear, and it contains some novel valuable results.

**Strength And Weaknesses:**

Strength:

[1] The problem of studying coordination and heterogeneity levels of cooperative MARL environments is interesting, and the developed new environments HECOGrid are helpful for better quantitative understanding of different MARL approaches across different levels of coordination and environmental heterogeneity, which could be valuable for future RL research.

[2] The proposed SAF method seems to be valid, and the provided empirical evaluations help demonstrate the advantages of the proposed SAF method.

[3] The paper is written reasonably clearly and I find it easy to follow.

Weakness/suggestions:

[1] As mentioned in the paper, environmental heterogeneity has been less explored in the literature. To better justify the significance of the contribution & better understand the motivation, I'd love to see some more explanations/discussions how important/essential for such separate concepts of "coordination" and "environmental heterogeneity" for MARL (rather than a single concept of "difficulty level" in some previous MARL environments, which implicitly supports various levels of  "coordination" and "environmental heterogeneity"), and why this is super important from MARL research perspective. Such explanation/clarification would be helpful to justify that the contribution here is significant enough.

[2] It would be valuable/helpful if the authors could elaborate a bit more about the supported scale & complexity for developed new environments HECOGrid, against previous MARL environments. Is  HECOGrid appropriate for all different scales (from small to super large) and complexities (from simple to super complicated) of MARL experiments, and how about the comparison with previous MARL environments? Table 1 shows that HECOGrid has some advantage over all the other previous MARL environments by enabling explicit control of "environmental heterogeneity", but I'd like to also see some more comprehensive/objective discussions on pros and cons of HECOGrid against other MARL environments. E.g., are there any scenarios where other MARL environments would be more appropriate/preferred for the experiments rather than HECOGrid? Is the main advantage for HECOGrid being that it allows explicit control of "environmental heterogeneity", and it'd be not that relevant for future research that is not focusing on "environmental heterogeneity"?

**Summary Of The Paper:**

In this paper, the authors explore coordination and heterogeneity levels of cooperative MARL environments. In particular, they formalize the notions of coordination level and heterogeneity level of an environment and present HECOGrid, a suite of multi-agent RL environments that facilitates empirical evaluation of different MARL approaches across different levels of coordination and environmental heterogeneity by providing a quantitative control over coordination and heterogeneity levels of the environment. They also proposed a new method named SAF (Stateful Active Faciliator) to enable agents to perform well in difficult environments with high levels of coordination and heterogeneity. They evaluated SAF and compare its performance against baselines IPPO and MAPPO on HECOGrid, and demonstrate the advantages of  the proposed SAF method.

**Summary Of The Review:**

The authors formalized two MARL concepts ("coordination" and "environmental heterogeneity"), developed new MARL environments HECOGrid incorporating these two concepts, and proposed a novel algorithm SAF whose advantages were demonstrated in empirical evaluations in HECOGrid. The proposed method is heuristic in nature, and the contribution is mainly on empirical side and minor from theoretical side.

I found the paper is generally written clear, and it does contain some valuable novel results which might be appropriate for publication on ICLR, though I'd like to see some more clarifications/explanations from the authors to better justify the contributions & motivation, as described in above "strength and weakness" section.

---

### Decision · Program_Chairs · 2023-01-20

**Decision:**

Accept: poster

**Justification For Why Not Higher Score:**

- Paper lacks focus at times and does not always clearly articulate arguments --- it tries to do too much
- Concern that the definition of coordination is not sufficiently general

**Justification For Why Not Lower Score:**

- Strong empirical evaluation
- Suite of environments seems quite useful
- Value in pulling out coordination and heterogeneity as distinct concepts

**Metareview: Summary, Strengths And Weaknesses:**

This paper considers coordination in heterogeneous multi-agent RL (MARL).  It formalizes these notions of coordination and heterogeneity and argues that heterogeneity has been understudied in MARL.  The paper presents HECOGrid, which is a new suite of MARL environments that allows varying the amount of heterogeneity faced across agents and the level of coordination required. It also presents a new method, Stateful Active Facilitator (SAF), designed for settings with high heterogeneity and a high level of coordination required.  SAF outperforms two baselines on HECOGrid.

Strengths

- Coordination and heterogeneity are both important in MARL

- The benchmark suite HECOGrid seems like a useful new way of quantifying performance while varying coordination requirements and heterogeneity

- It seems useful to pull out heterogeneity and coordination level as specific problem characteristics and to be able to study them differently from other aspects of an environment's difficulty

- Strong empirical evaluation

Weaknesses

- The definition of the coordination level considers each time step in isolation. This does not allow capturing coordination that is temporal, where one agent needs to take an action after another agent takes its action.

- It is possible that this paper is trying to do too much, (tackling coordination, heterogeneity, and both providing an algorithm and a new suite of environments) as articulated by reviewers y8q8 and PAoT and as is evident by some of the details left out articulated by reviewer UJ6m. It seems possible to do this, but it will require the appendix to be longer than usual and will put a burden on the authors to be especially clear and organized.

- Opportunities to improve clarity and motivation, as called out in the individual reviews from all reviewers

Other comments

- Initial experiments used observations rather than histories as the input to agents. Follow-up experiments addressed this, using RNNs that take the history as input. Additional experiments also added new baselines.

**Note From Pc:**

if the above contains the word "oral" or "spotlight" please see: "oral" presentation means -> notable-top-5% and "spotlight" means -> notable-top-25%. As stated in our emails, we are disassociating presentation type from AC recommendations

**Summary Of Ac-Reviewer Meeting:**

While the ratings for this paper were divergent, this was not considered a borderline paper as 4 other paper with borderline scores and divergent ratings were prioritized for AC-reviewer meetings, and as the original ratings put it *below* the 5.25 cutoff. It is only in taking a more detailed pass through the reviews and paper that I realize that we should discount the review with a rating of 3.

I have emailed the two reviewers with the lowest scores to ask them to take a look at the author rebuttals and other reviews to see if it updates their views on the paper. The rating of a 3 from one reviewer is inappropriately low, given that all but one concern was minor as I see it, and the authors addressed the one major concern. The rating of 5 from the other reviewer is a reasonable viewpoint, but that reviewer has nonetheless not acknowledged the rebuttal or other reviews.

UPDATE AFTER ORIGINALLY SUBMITTING METAREVIEW
The two reviewers with lower scores have updated their scores to a 6 after reading the other reviews and the author rebuttal.